# Surveying a Sample of the Spanish Ophthalmologic Community about Vaccination against Herpes Zoster

**DOI:** 10.3390/vaccines11050952

**Published:** 2023-05-05

**Authors:** Alberto Sánchez-Mellado, Luis Alcaraz-Clemente, Marina Rodríguez-Calvo-de-Mora, José-María Sánchez-González, Santiago Ortiz-Perez, Carlos Rocha-de-Lossada

**Affiliations:** 1Hospital Universitario Virgen de las Nieves, Av. de las Fuerzas Armadas, 2, 18014 Granada, Spain; 2Hospital Regional Universitario de Málaga, Plaza del Hospital Civil, S/N, 29009 Malaga, Spain; 3Qvision, Department of Ophthalmology of VITHAS Almería Hospital, 04120 Almería, Spain; 4Ophthalmology Department, VITHAS Málaga, 29016 Malaga, Spain; 5Department of Physics of Condensed Matter, Optics Area, University of Seville, 41012 Seville, Spain; 6Surgery Department, Ophthalmology Area, University of Seville, 41013 Seville, Spain

**Keywords:** varicella zoster virus, Spanish ophthalmologist, survey, vaccination, varicella herpes zoster

## Abstract

There are currently two authorized vaccines against herpes zoster (HZ) that have been shown to be safe and effective in its prevention: Zostavax, a zoster vaccine live (ZVL), and Shingrix, a recombinant zoster Vaccine (RZV). Because ophthalmologists work with vision-threatening complications of zoster, such as herpes zoster ophthalmicus (HZO), they are in a good position to advocate for vaccination. Our aim was to determine the current knowledge among Spanish ophthalmologists about the effectiveness of the available vaccines against HZ. A Google Forms questionnaire was created and used as the survey platform for this study. It was an anonymous online survey of 16 questions, which was shared among Spanish ophthalmologists in-training and consultants from 27 April 2022 to 25 May 2022. The survey was completed by a total of 206 ophthalmologists of all subspecialties. We obtained responses from 17 of the 19 regions of Spain. Fifty-five percent of the respondents agreed that HZ is a frequent cause of vision loss. However, 27% of the professionals were unaware of the existence of vaccines against HZ and 71% of them did not know in which cases it could be indicated. Only nine ophthalmologists (4%) had ever suggested vaccination against HZ to their patients. Despite this, 93% considered it important to recommend vaccination against HZ if it proved to be safe and effective. Considering the sequelae, complications, and the existence of safe and effective vaccines against HZ, vaccination of the target population could be considered an important public health measure. We are convinced that it is time for ophthalmologists to take an active role in HZO prevention.

## 1. Introduction

The varicella zoster virus (VZV) is a ubiquitous virus that causes varicella (chicken pox) as a primary infection and herpes zoster (HZ) (shingles) when a latent virus reactivation occurs [1]. The lifetime risk of developing HZ for an individual is approximately 30% [2]. The incidence rate increases significantly with age and is currently on the rise around the world [3,4,5]. HZ is frequently diagnosed in Spain, with an estimated annual incidence of 351.6 cases per 100,000 inhabitants [6].

HZ manifests itself as a unilateral, vesicular, painful rash in a single dermatomal distribution, usually involving the thoracic and lumbar dermatomes. Since VZV can remain dormant in any ganglion, its manifestations can vary greatly, from paraparesis (vertebral ganglia) to motor neuropathy (sensory ganglia) [1,2]. HZ can lead to several complications such as stroke, persistent pain three months after the onset of the rash (known as post-herpetic neuralgia, PHN), cranial nerve palsies, or herpes zoster ophthalmicus (HZO) [7], which occurs as a consequence of VZV reactivation involving the ophthalmic branch of the trigeminal nerve.

HZO rates are currently increasing [8,9]. Approximately 10–20% of cases of HZ present as HZO [2], but this frequency may be higher. In a cohort of patients from Minnesota, an overall 23% increase in ocular complications of HZ was found between 1980 and 2007 [10]. There are several hypotheses that may justify the increase in the number of cases of HZ/HZO, including the facts that (a) chickenpox immunisation among children has led to less exposure to small amounts of live virus among young adults, which normally would have boosted their immunity, (b) those immunised against chickenpox who are now older have weaker immunity compared to infection with the “wild-type” strain, and so latent strains may be more able to reactivate, and (c) immunosuppressed individuals are at greater risk of reactivation and (d) the introduction of vaccines for HZ [8,11]. These reasons may explain why most new cases of HZO occur in the population aged 31 to 60 years [8,11].

HZO complications include preseptal cellulitis, keratitis, corneal scarring, glaucoma, cataract, uveitis, retinitis, choroiditis, optic neuritis, and permanent vision loss [1,10]. These complications can present in an acute, chronic, or recurrent form. Therefore, VZV infection is an important cause of vision loss, pain, morbidity, and decreased quality of life [8].

Cellular immunity keeps VZV in a latent state in the ganglia [6]. Cell-mediated immunity increases after VZV infection, but this protection wears off over time [6]. Therefore, active immunity measures, such as vaccination, are needed.

Vaccination may be the key to prevent HZ disease and its possible complications [12,13]. There are two HZ vaccines currently available on the market: Zostavax (Merck & Co, Inc), a zoster vaccine live (ZVL) [14], and Shingrix (GlaxoSmithKline), a recombinant zoster vaccine (RZV). Both vaccines are effective in reducing the incidence of HZ (51.3% and 96.2%, respectively) and its complications [13]. Moreover, RZV has also demonstrated efficacy in a real-world approach [15]. In Spain, vaccination is recommended for the prevention of HZ and PHN in adults over 50 years of age [16]. Given the significant burden of this disease, low vaccination rates are alarming (approximately 65% of adults recommended to receive this vaccine remain unprotected) [17,18]. Given the frequency of ocular involvement in HZ, ophthalmologists are well positioned to have the opportunity to educate patients regarding the benefits of HZ vaccination [2,13,19].

Studies such as the work Galvis V. et al. have already analysed the importance of vaccination against HZ [20]. Surveys such as the one by Tsui E. et al. have been conducted to investigate the knowledge, attitudes, and practice patterns of primary care physicians regarding the administration of herpes zoster vaccination [21].

The aim of this research survey was to assess the knowledge about HZ vaccines among a sample of ophthalmologists in our country.

## 2. Materials and Methods

To perform this study, we started with the following research question according to the PICO framework [22]. Is it known among Spanish ophthalmologists/trainees that the population vaccinated with HZ has a lower rate of ophthalmological complications compared to the unvaccinated population?

There was no previously validated questionnaire. Therefore, we designed a questionnaire based on the aim of this research and the published literature. A Google Forms questionnaire was created and used as the survey platform for this study. It was an anonymous online survey that was shared among Spanish ophthalmologist trainees and consultants over a period from 27 April 2022 to 25 May 2022.

The first draft of the questionnaire was created by two of the authors (AS and LA) after performing a literature review, which included the most relevant articles related to this topic. Specifically, we reviewed the existing bibliography on MEDLINE, Pubmed, and the EQUATOR (Enhancing the QUAlity and Transparency Of health Research) Network’s websites. In particular, four of the articles were very helpful for this purpose [23,24,25,26]. For new questions, we used the acronym BRUSO (brief, relevant, unambiguous, specific, and objective) developed by Peterson [27]. After the literature research, the senior authors (CRL, MRCM, and SOP) reviewed and modified the questionnaire until all authors were in agreement. The questions used in the questionnaire were as follows:Select your professional category.Select the subspecialty to which you dedicate most of your time.Professional experience.Work center where you carry out most of your activity.Autonomous Community in which you carry out most of your work activity.How often do you see patients who have suffered or are suffering from ophthalmological involvement of HZ?Do you think that HZ is a common cause of vision loss?What percentage of HZ disease do you consider has ophthalmological involvement?Do you know about the existence of vaccines for HZ?Do you know if there is an HZ vaccine marketed in Spain?If so, do you know in which cases vaccination is recommended?Have you ever indicated vaccination against HZ to any of your patients?If you want to recommend vaccination against HZ, do you know how to request it?How effective is vaccination with Shingrix (recombinant vaccine) in the prevention of ophthalmicus HZ?How effective is vaccination with Zostavax (attenuated virus vaccine) in the prevention of ophthalmicus HZ?If vaccination against HZ were effective and safe, do you think it would be important to recommend it from an ophthalmological point of view?

The sampling frame was determined through non-random convenience sampling with the aim of being administered to the most readily available ophthalmologists around Spain [23]. The inclusion criteria to fill in the questionnaire were being an ophthalmologist consultant or an ophthalmology resident working in Spain at the moment of the survey. The survey was sent by email or WhatsApp^®^ (Meta Platforms, Inc., Menlo Park, CA, USA) to the population sample frame. The distribution of the survey was carried out through email and/or the WhatsApp^®^ application using private groups of ophthalmologists from the main hospitals in Spain in each region. The emails and telephone numbers were obtained through the head of service of our hospital. In a similar way, to obtain a sample of the different subspecialties, the survey was distributed through WhatsApp groups of specific subspecialties (i.e., cornea, glaucoma, cataract and refractive, etc).

As a quality control before distribution, we carried out a pilot test of the questionnaire. We asked a group of ophthalmologists different from the current authors to review the survey and identify any latent errors. Pilot tester responses were excluded in the subsequent data analysis. Responses were anonymous and confidential. These ophthalmologists were asked about their knowledge of HZ and the existence of vaccines. They were also asked about the efficacy of these vaccines, their recommendation, and the way in which they should be requested. Finally, as previously noted, we shared the questionnaire with Spanish ophthalmologists/trainees in a non-random convenience sample. The survey was completely anonymously, and the data derived from it were treated with strict confidentiality. We conducted a survey among Spanish ophthalmologists and trainees, using a non-random convenience sampling method. The survey questionnaire was developed in Spanish and included the medical terms ‘herpes zoster’, ‘herpes zoster ophthalmicus’, and ‘shingles’ translated into Spanish as ‘herpes zóster’, ‘herpes zóster oftálmico’, and ‘culebrilla’, respectively. We did not include an English version of the survey, as it was designed specifically for the Spanish ophthalmologic community. The survey was completely anonymous, and the data derived from it were treated with strict confidentiality.

### Statistical Analysis

Statistical analysis was carried out using IBM SPSS Statistics, version 25.0 (IBM Corp., Armonk, NY, USA, 2017). All variables included in the study were qualitative and expressed as absolute frequency and percentage. The chi-square test and Fisher’s exact test were used in the study of the association between variables. Correlation was assessed by calculating Spearman’s correlation coefficient (ρ). A *p* value < 0.05 was considered statistically significant.

## 3. Results

The survey was completed by a total of 206 ophthalmology professionals. All of them were included in the statistical analysis as no erroneous answers were recorded. Of the 19 regions that make up our country, all were represented except Cantabria and La Rioja (two small regions that have fewer than one million inhabitants combined). The region with the highest participation was Andalusia, with 122 responses (59%), followed by the Community of Madrid, with 24 (12%), and Cataluña, with 22 (11%) (these are the three most-populated regions in Spain and account for almost 50% of the population).

The vast majority of respondents, 177, were qualified ophthalmology specialists (86%), and the remaining percentage were ophthalmology residents (14%). With regard to the distribution of sections, 45 subjects were dedicated to general ophthalmology (22%), followed by cornea and ocular surface (20%) and retina (17%) (Figure 1). More than 30% of the professionals had more than 20 years of experience, which provided valuable information on the perception of ophthalmologic complications due to zoster over time. Approximately 50% practiced exclusively in public centres, while 28% combined public and private practice.

Regarding the questions directly related to HZ after excluding ophthalmology residents, we found that 54% of the professionals described monthly contact with patients with ophthalmologic involvement of HZ. Only 8% reported weekly contact. These percentages were very similar among the different subspecialties, with some exceptions. In the case of paediatric ophthalmology, most of the respondents (69%) described contact with these patients on an annual basis. For cornea and ocular surface specialists, 18% saw this type of pathology every week. Of the total respondents, 95 (54%) agree that HZ is a frequent cause of vision loss. When performing an analysis by sections, similar figures were found in the different subspecialties.

As shown in Table 1, we found a significant low negative correlation between professional experience and the belief that VZV is a frequent cause of vision loss. The greater the professional experience, the lower the percentage of visual loss attributed to HZ (ρ = −0.22, *p* = 0.003). Specifically, 74% of ophthalmologists with less than 5 years of experience thought that HZ is a frequent cause of visual loss, compared to 58% for the group with 5–10 years of experience, 51% for 10–20 years of experience, and 43% for those with more than 20 years of experience, χ2(3, N = 175) = 9.39, *p* = 0.025 (see Table 1).

Respondents were asked what fraction of patients with HZ developed ocular involvement. Most (43%) thought the eye was affected in 10–20% of patients with zoster; 29% thought ocular involvement occurred in <10% of those with zoster. A similar distribution was found in the cornea and ocular surface subgroup.

The results about immunization against HZ are shown in Table 2. In addition to the whole sample, an analysis that excludes residents and another that includes only the corneal and ocular surface subgroups was performed and is presented in the second and third columns, respectively. Overall, 26.7% of all respondents were unaware of the existence of vaccines. Half of those surveyed did not know that vaccination was marketed in Spain. Furthermore, 70.9% did not know in which cases it could be recommended. In the event that its use would be recommended, 90.3% did not know how to request the vaccine. This circumstance was especially prevalent among ophthalmology residents, where the figure reached 100%. Only nine ophthalmologists (4%) had ever recommended vaccination against HZ to their patients. In this regard, neither professional experience nor the development of the activity in a public or private centre was a differentiating factor.

Regarding the efficacy rate of the Shingrix (RZV) and Zostavax (ZVL) vaccines (Table 3), most respondents were unaware of the superiority of Shingrix over Zostavax. They considered both to be 50–80% effective. However, when we analysed only the cornea and ocular surface group specialists, they had a slightly higher non-significant difference regarding the efficacy (50–80% and >80% effectiveness) of HZ vaccines when compared to the rest of the ophthalmologists (Table 3).

Finally, 93% considered it important to recommend vaccination against HZ if it was proved to be effective and safe. In the subgroup of more experienced ophthalmologists, this percentage was 84% (χ2 (6, N = 206) = 12.1, *p* = 0.07). In this subgroup, five ophthalmologists (two of them subspecialists in cornea) answered that they would extend the recommendation of vaccination in situations of recurrence and immunosuppression.

## 4. Discussion

HZ and HZO are frequent entities of growing importance in developed societies [6]. As immunity produced after primary VZV infection does not protect against HZ, passive immunity measures such as vaccination are needed.

In Spain, most of the ophthalmologists saw patients with HZO on a monthly basis and considered it a frequent cause of vision loss. It is remarkable that most ophthalmologists who answered the questionnaire (93%) would recommend vaccination if it was proved to be safe and effective, but very few had prescribed it in their practice (4%). Among the causes could be a lack of information, as 71% did not know the indications for the vaccine and 90% did not know how to request it. In the near future the difficulties they have experienced in prescribing vaccination could be investigated.

Despite the theoretical interest in the vaccine, real knowledge of vaccination and its importance in the prevention of complications were poor. As highlighted by Virgilio Galvis and colleagues, ophthalmologists may be less aware of the need for and effectiveness of the HZ vaccine than general practitioners [20]. However, if we compare the results of our study with data from a survey of 138 physicians at an American institution (New York University, Langone Health), we can see that the results for both specialties are similar. In general, 91% of primary care physicians recommended that all immunocompetent patients 60 years of age or older should be vaccinated, compared to 93% of the ophthalmologists who answered the survey in our country and would recommend vaccination [21]. This contrasts with the low percentage of our respondents who have ever recommended the vaccine to their patients (4%). A lack of knowledge about the vaccine is also widespread in the general population. A survey of the general public in our country found that only 10% of participants knew that an HZ vaccine existed [28]. One of the reasons that may explain these differences is the number of awareness campaigns that have been carried out in the United States, as well as the consideration of VZV as a systemic disease among general practitioners. Healthcare provider recommendations for vaccination are strongly associated with a patient’s receipt of vaccines [29].

Vaccination rates against HZ are generally poorly reported. In some countries such as the United Kingdom and the United States, vaccination rates range from 24.1 to 72% depending on the age group analysed [18,30]. In Spain, it is difficult to find vaccination rates for HZ in the general population. This may be due to the fact that it was not included in the vaccination schedule recommended by the Interterritorial Council of the National Health System until 2022 [31]. Previously, in 2018, vaccination with the recombinant formulation was recommended in patients ≥50 years of age with various situations that increase the risk of the disease (haematopoietic progenitor cell transplantation, solid organ transplantation, treatment with anti-JAK drugs, HIV, haematological malignancies, and solid tumours undergoing chemotherapy) [31]. Moreover, it is important to note that serious adverse events (SAEs) have been reported with both the RZV and ZVL vaccines. In fact, three adults who were given ZVL (Zostavax) died from the disseminated live vaccine virus. Two of the adults had cancer (CLL) and one had rheumatoid arthritis [32]. The restrictive indications of the Interterritorial Council of the National Health System before 2022 are surprising in view of the high effectiveness and safety of the HZ vaccines, particularly the RZV [12,33].

Regarding the economic analysis of vaccination, although there is significant heterogeneity, most studies determined that HZ vaccination was cost-effective when given at ages over 65 or 70 [34]. In a study specifically conducted in Spain, it was shown that the most cost-effective strategy in the general population was vaccination of the 65-year-old and older cohort with the RZV (Shingrix). [35]

Our study has limitations; first, this is a survey study in which the outcomes depend on the willingness of respondents to respond appropriately. Moreover, convenience sampling may imply a bias when considering that the results from this survey may be extrapolated to the entire community of ophthalmologists, since the way in which the survey was distributed by e-mail and WhatsApp^®®^ groups means that we do not know precisely the response rate among those who received the survey. Nonetheless, this is a common way of conducting this type of study [36,37]. The questionnaire with closed answers is also another limitation that should be mentioned, since respondents can provide relevant information in open answers. All of these limitations should be taken into account for new studies. However, regarding the strength, this is a survey with a high percentage of responses among the community of ophthalmologists in Spain, which reaffirms the outcomes obtained a few years ago by Tsui et al. [21].

## 5. Conclusions

In conclusion, we have detected that there is a lack of knowledge about the effectiveness of vaccination against HZ in the prevention of ophthalmological complications among the ophthalmologists surveyed, in addition to low rates of recommendation and knowledge on how to request the vaccine. Higher vaccination rates in the near future will only be possible with the awareness and action of all physicians, both primary care and specialists. HZ vaccine recommendation training programs could be carried out in Spain. We truly believe that the time has come for ophthalmologists to take an active role in HZO prevention.

## Figures and Tables

**Figure 1 vaccines-11-00952-f001:**
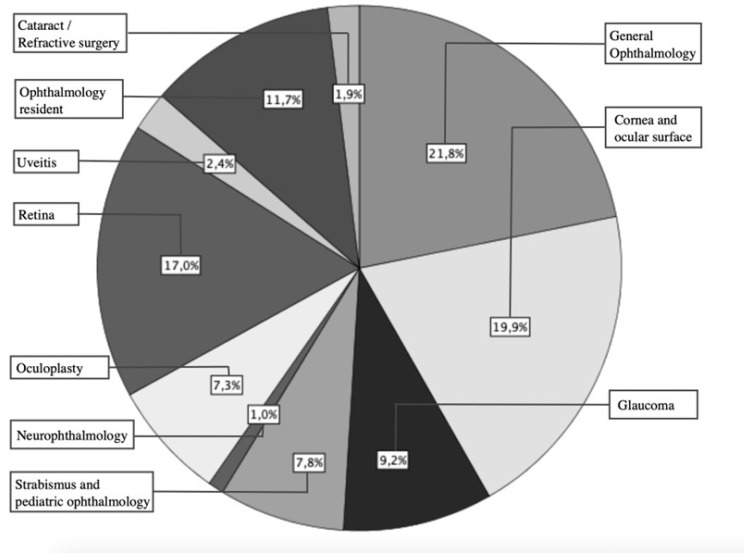
Subspecialty distribution of the participants.

**Table 1 vaccines-11-00952-t001:** Percentage of ophthalmologists who considered that HZ is a common cause of visual loss according to years of professional experience.

	Years of Experience				
	<5	5–10	10–20	>20	*p* Value
Percentage of ophthalmologists who considered that HZ is a common cause of visual loss n (%)	29 (74%)	18 (58%)	21(51%)	28(43%)	*p* = 0.003

**Table 2 vaccines-11-00952-t002:** Knowledge of the participants about the existence, commercialization, indication, and request of the HZ vaccines.

Question	“Yes” Respondents n (%)		
	All Respondents	Respondents Excluding Trainees	Cornea and Ocular Surface Subgroup
Do you know about the existence of vaccines for HZ?	151 (73.3)	127 (72.6)	35 (89.7)
Do you know if there is a HZ vaccine marketed in Spain?	104 (50.5)	89 (50.9)	24 (61.5)
Do you know in which case vaccination is indicated?	60 (29.1)	54 (30.9)	16 (41)
If you want to recommend vaccination against HZ, do you know how to request it?	20 (9.7)	19 (10.9)	5 (12.8)

**Table 3 vaccines-11-00952-t003:** Efficacy rate estimation of the Shingrix© (GlaxoSmithKline, Brentford, United Kingdom) and Zostavax© (Merck, Darmstadt, Germany) vaccines.

Efficacy Rate Estimation of the Shingrix© and Zostavax© Vaccines	All Respondents n(%)		Cornea and Ocular Surface Subgroup n(%)	
	Shingrix©	Zostavax©	Shingrix©	Zostavax©
<20%	9 (4.4)	14 (6.8)	2 (4.9)	3 (7.3)
20–50%	69 (33.5)	70 (34)	10 (24.4)	10 (24.4)
50–80%	95 (41.6)	98 (47.6)	21 (51.2)	21 (51.2)
>80%	33 (16)	24 (11.7)	8 (19.5)	7 (17.1)

## Data Availability

Data available on request due to restrictions. The data presented in this study are available on request from the corresponding author.

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
