# Peer review of "Surveying a Sample of the Spanish Ophthalmologic Community about Vaccination against Herpes Zoster"

_vaccines, 2023, doi:10.3390/vaccines11050952_

Round 1

Reviewer 1 Report (Previous Reviewer 3)

No further comments on the content of the manuscript.

There are still some minor edits required.  The title is not grammatically correct.  There are errors in the first sentence of the abstract.  There's a spelling mistake at the end of line 56; "stains" should be "strains".  Please check the manuscript thoroughly for further grammatical and spelling errors.

Author Response

Reviewer 1

Comments and Suggestions for Authors

#RV1: No further comments on the content of the manuscript.

#AU1: Many thanks for your comments

Comments on the Quality of English Language

#RV2: There are still some minor edits required.  The title is not grammatically correct. 

#AU2: Thanks for the comment, correction made.

#RV3: There are errors in the first sentence of the abstract. 

#AU3:  Thanks for the comment, correction made.

#RV4: There's a spelling mistake at the end of line 56; "stains" should be "strains".  Please check the manuscript thoroughly for further grammatical and spelling errors.

#AU4: Thanks for the comment, correction made.

Reviewer 2 Report (Previous Reviewer 2)

This revised version is improved.

Two minor suggestions:

1. First sentence of Abstract needs editing/

2. P valor in Table 1 should be P value

Some minor corrections suggested above.

Author Response

Reviewer 2

Comments and Suggestions for Authors

#RV0: This revised version is improved.

#AU0: Many thanks for your comment.

Two minor suggestions:

#RV1: First sentence of Abstract needs editing

#AU1: Thanks for the comment, correction made.

#RV2. P valor in Table 1 should be P value

#AU2: Thanks for the comment, typo error.

Comments on the Quality of English Language

#RV3: Some minor corrections suggested above.

#AU3: Thanks for the comment, all correction improved.

Reviewer 3 Report (New Reviewer)

These authors from Spain propose that ophthalmologists could take a more active role in promoting vaccination to prevent herpes zoster.  They outline their strategy in the manuscript.  This Reviewer agrees with their proposal.  Therefore, this manuscript fits well within the journal VACCINES.  A few comments for improvement are listed below.

1.  Introduction, line 71.  First article about the Zoster Vaccine Live, also called Zostavax.  There is one 2005 reference about the clinical study of ZVL that has been cited almost 3000 times and is known around the world.  Suggest that the authors add the reference by M. Oxman et al, New England Journal of Medicine 352: 2271-84, 2005.  PMID:15930418.

2.  Methods, section 2 about the survey, line 144.  Many of the readers of this journal will be bilingual or trilingual.  Also, other investigators may want to perform a similar study in a Spanish speaking country.  Therefore, please add a few more sentences.  Explain if the survey included both Spanish and English languages?  Were any other languages included? In English, the important medical words would be (1) herpes zoster, (2) herpes zoster ophthalmicus and (3) shingles.  Explain what words were used in the Spanish version for these 3 English names.

3.  Results.  Figure 3 provides good insight into lack of basic knowledge about zoster vaccines among doctors in Spain.  Nothing more to do. 

4.  Discussion, line 254.  Please add a very important reference about serious adverse events at this location.  The sentence on lines 250-254 has very relevant medical data about RZV.  After reference 32, add a comment that 3 adults who were given ZVL (Zostavax) have died from disseminated live vaccine virus.  Two of the adults had cancer (CLL) and one had rheumatoid arthritis.  The three patients were from Canada, England and Australia.  Here is the reference: N. Price et al, VACCINES (Basel)9:23, 2021.  See Table 2. PMID:33418856.

5.  Reference section, line 300.  Add 2 new references.

Author Response

Reviewer 3

#RV0: These authors from Spain propose that ophthalmologists could take a more active role in promoting vaccination to prevent herpes zoster.  They outline their strategy in the manuscript.  This Reviewer agrees with their proposal.  Therefore, this manuscript fits well within the journal VACCINES.  A few comments for improvement are listed below.

#AU0: Many thanks for your time and revision in the manuscript.

#RV1: Introduction, line 71.  First article about the Zoster Vaccine Live, also called Zostavax.  There is one 2005 reference about the clinical study of ZVL that has been cited almost 3000 times and is known around the world.  Suggest that the authors add the reference by M. Oxman et al, New England Journal of Medicine 352: 2271-84, 2005.  PMID:15930418.

#AU1: We agree with your comment, and we have included the reference [1].

#RV2: Methods, section 2 about the survey, line 144.  Many of the readers of this journal will be bilingual or trilingual.  Also, other investigators may want to perform a similar study in a Spanish speaking country.  Therefore, please add a few more sentences.  Explain if the survey included both Spanish and English languages?  Were any other languages included? In English, the important medical words would be (1) herpes zoster, (2) herpes zoster ophthalmicus and (3) shingles.  Explain what words were used in the Spanish version for these 3 English names.

#AU2: Thanks for the comment, we conducted a survey among Spanish ophthalmologists and trainees, using a nonrandom convenience sampling method. The survey questionnaire was developed in Spanish and included the medical terms 'herpes zoster,' 'herpes zoster ophthalmicus,' and 'shingles' translated into Spanish as 'herpes zóster,' 'herpes zóster oftálmico,' and 'culebrilla,' respectively. We did not include an English version of the survey, as it was designed specifically for the Spanish ophthalmologic community. The survey was completely anonymous, and the data derived from it were treated with strict confidentiality.

#RV3: Results.  Figure 3 provides good insight into lack of basic knowledge about zoster vaccines among doctors in Spain.  Nothing more to do.

#AU3: Many thanks for your comments.

#RV4.  Discussion, line 254.  Please add a very important reference about serious adverse events at this location.  The sentence on lines 250-254 has very relevant medical data about RZV.  After reference 32, add a comment that 3 adults who were given ZVL (Zostavax) have died from disseminated live vaccine virus.  Two of the adults had cancer (CLL) and one had rheumatoid arthritis.  The three patients were from Canada, England and Australia.  Here is the reference: N. Price et al, VACCINES (Basel)9:23, 2021.  See Table 2. PMID:33418856.

#AU4: Moreover, it is important to note that serious adverse events (SAEs) have been reported with both RZV and ZVL vaccines. In fact, three adults who were given ZVL (Zostavax) have died from disseminated live vaccine virus. Two of the adults had cancer (CLL) and one had rheumatoid arthritis [2].

#RV5.  Reference section, line 300.  Add 2 new references.

#AU5: Both references were included in the references section.

Reference

  1. Oxman, M.N.; Levin, M.J.; Johnson, G.R.; Schmader, K.E.; Straus, S.E.; Gelb, L.D.; Arbeit, R.D.; Simberkoff, M.S.; Gershon, A.A.; Davis, L.E.; et al. A Vaccine to Prevent Herpes Zoster and Postherpetic Neuralgia in Older Adults. N. Engl. J. Med. 2005, 352, 2271–2284, doi:10.1056/NEJMOA051016.
  2. Price, N.B.; Grose, C. Corticosteroids Contribute to Serious Adverse Events Following Live Attenuated Varicella Vaccination and Live Attenuated Zoster Vaccination. Vaccines 2021, 9, 1–12, doi:10.3390/VACCINES9010023.

This manuscript is a resubmission of an earlier submission. The following is a list of the peer review reports and author responses from that submission.

Round 1

Reviewer 1 Report

            Alberto Sánchez-Mellado, et al. surveyed Spanish ophthalmologists regarding their knowledge of VZV vaccines. Two hundred and six ophthalmologists from 17 regions answered. Fifty-five % of them answered that VZV is the main cause of vision loss. However, 22% of them did not know about the existence of VZV vaccines. Ninety-three % of them recommended that VZV vaccines should be immunized if these is safe and effective. The authors also concluded that ophthalmologists should recommend VZV vaccination.

General comments:

              From contents of this paper, it is understandable that the authors thought that ophthalmologists should also recommend VZV vaccines. However, the content should be submitted to ophthalmological journals.

If vaccines are described as VZV vaccines, it is perceived to include an attenuated live varicella vaccine. If the authors are discussing HZ, they should be unified on HZ vaccines.

Minor comment

Page1, Line 40: HZ infection should be HZ only.

Method:Should the name of Google Forms be used in scientific journals?

Page 2, Line 91-99: This description seems unnecessary. It should simply state what survey items were developed based on the references.

Page 2, Line 100-101: The text is also unnecessary. It is sufficient to purely describe the items surveyed.

Page 2, Line 105 – 111: Is there any official confirmation that they are ophthalmologist?

Page 2, Line 113 – 121: Is the pilot study worthwhile?

Result

Page 4, Line 156 – 165: Can you summarize this paragraph in a figure or table? It is difficult to understand the text alone.

Author Response

Reviewer 1

#RV0: Alberto Sánchez-Mellado, et al. surveyed Spanish ophthalmologists regarding their knowledge of VZV vaccines. Two hundred and six ophthalmologists from 17 regions answered. Fifty-five % of them answered that VZV is the main cause of vision loss. However, 22% of them did not know about the existence of VZV vaccines. Ninety-three % of them recommended that VZV vaccines should be immunized if these is safe and effective. The authors also concluded that ophthalmologists should recommend VZV vaccination.

#AU0:

General comments:

 #RV1: From contents of this paper, it is understandable that the authors thought that ophthalmologists should also recommend VZV vaccines. However, the content should be submitted to ophthalmological journals. If vaccines are described as VZV vaccines, it is perceived to include an attenuated live varicella vaccine. If the authors are discussing HZ, they should be unified on HZ vaccines.

#AU1: We have decided to submit this article to this prestigious journal for wider dissemination, not only within ophthalmology. In the introduction section we described the two types of VZV vaccines referred to throughout the paper (Shingrix and Zostavax). Thank you for your kind comment, however in this paper we do not focus on the role of the live attenuated virus vaccine used in the prevention of varicella (chickenpox) and just in Shingrix and Zostavax.

Minor comment

#RV2: Page1, Line 40: HZ infection should be HZ only.

#AU2: We have deleted the word “infection”.

#RV3: Method:Should the name of Google Forms be used in scientific journals?

#AU3:  Google Forms is a tool with a format that is familiar to the population and easy to use, which is why we chose this platform to distribute the survey. In the literature, this platform has also been selected in other articles already published in scientific journals.

#RV4: Page 2, Line 91-99: This description seems unnecessary. It should simply state what survey items were developed based on the references.

#AU4: As there was no previously validated survey, we have detailed the steps we followed to construct our survey. If it is considered unnecessary we can narrow it down.

#RV5: Page 2, Line 100-101: The text is also unnecessary. It is sufficient to purely describe the items surveyed.

#AU5: We make the suggested change.

#RV6: Page 2, Line 105 – 111: Is there any official confirmation that they are ophthalmologist?

#AU6:  We can guarantee that the survey has only been sent to ophthalmologists, but what we cannot guarantee, as in all surveys, is an official confirmation from each of the ophthalmologists.

#RV7: Page 2, Line 113 – 121: Is the pilot study worthwhile?

#AU7: We believe it is worthwhile because the analysis of third parties as an example of a real population may show errors that had not been recognized. Actually, we follow standard surveys methodologies papers cited in this article.

Result

#RV9: Page 4, Line 156 – 165: Can you summarize this paragraph in a figure or table? It is difficult to understand the text alone.

#AU9: We have summarized this data in the following table for better understanding:

Years of experience

<5

5-10

10-20

>20

p valor

Percentage of ophthalmologists who considered that VZV is a common cause of visual loss. n (%)

29 (74%)

18 (58%)

21(51%)

28(43%)

p=0.003

Table 1. Percentage of ophthalmologists who consider that VZV is common cause of visual loss according to the years of professional experience.

Reviewer 2 Report

Reviewer comments

The article by Sanchez-Mellado et al. describes a questionnaire survey of ophthalmologists in Spain regarding their knowledge about Zoster vaccines and whether they prescribe a Zoster vaccine. The survey results are somewhat surprising in how much room there is for improvement in educating ophthalmologist about public health vaccination measures they can implement to prevent an important cause of eye disease. The paper is important in that it may help enlist ophthalmologists globally as vaccine allies to recommend Zoster vaccination in their patients.

I have one important suggestion and several minor edits.

Important suggestion:

1.     I suggest that the full questionnaire be published along with the paper, either by expanding the current table 1, or as a supplement. Without the questionnaire, no one will be able to repeat the study and it is more difficult to fully judge the quality of the data without access to the questionnaire.

Minor comments:

1.     Fig 2 is not easy to follow. Perhaps it is better as a table. There is a typo in the first column, black shading: 9 (44%) should be 9 (4,4%).

2.     Introduction: I think reference 26, Tsiu et al belongs in the Introduction to tell the reader about other surveys that have been done. Perhaps same for reference 25.

3.     Introduction lines 57-58: the statement “d) the introduction of vaccines for HZ (few reports on the influence of these vaccines on HZ rates have been published [8,11].” Consider modifying this statement or perhaps clarifying the intent of the statement, Several excellent publications in NEJM have described the remarkable efficacy of the Shingrix vaccine in various age groups.

4.     Introduction line 67: “Thus passive immunity measures such as vaccination are needed”. I suggest deleting “passive”. I do not think vaccination with attenuated live virus or subunit proteins is “passive” immunity.

5.     Some typos or grammar edits:

a.     Line 159: change “through” to “thought”.

b.     Line 170: change “for vaccination” to “that vaccination”.  

c.     Line 185: change “presuppose” to “had”.

d.     Lines 195-197: consider deleting “on a case-by-case basis, especially” from the sentence.

e.     Line 200: Consider changing “antibodies“ to “immunity” .... does not protect.

f.      Line 202: delete “we”.

g.     Line 241: Consider changing “This study” to “Our study”.

h.     Line 242: Place a period, not a comma, after “appropriately”.

Author Response

Reviewer 2

#RV0: The article by Sanchez-Mellado et al. describes a questionnaire survey of ophthalmologists in Spain regarding their knowledge about Zoster vaccines and whether they prescribe a Zoster vaccine. The survey results are somewhat surprising in how much room there is for improvement in educating ophthalmologist about public health vaccination measures they can implement to prevent an important cause of eye disease. The paper is important in that it may help enlist ophthalmologists globally as vaccine allies to recommend Zoster vaccination in their patients.

#AU0:

#RV1: I have one important suggestion and several minor edits.

Important suggestion:

  1. I suggest that the full questionnaire be published along with the paper, either by expanding the current table 1, or as a supplement. Without the questionnaire, no one will be able to repeat the study and it is more difficult to fully judge the quality of the data without access to the questionnaire.

The survey is attached as an appendix at the end of the manuscript.

Minor comments:

#RV2: Fig 2 is not easy to follow. Perhaps it is better as a table. There is a typo in the first column, black shading: 9 (44%) should be 9 (4,4%).

#AU2: We have replaced Fig 2 for Table 3 and corrected the previous typo in the first column, as follow:

Efficacy rate estimation of the Shingrix© and Zostavax© vaccines

All respondents

n(%)

Cornea and Ocular surface subgroup

n(%)

Shingrix©

Zostavax©

Shingrix©

Zostavax©

<20%

9 (4.4)

14 (6.8)

2 (4.9)

3 (7.3)

20 – 50%

69 (33.5)

70 (34)

10 (24.4)

10 (24.4)

50 – 80%

95 (41.6)

98 (47.6)

21 (51.2)

21 (51.2)

>80%

33 (16)

24 (11.7)

8 (19.5)

7 (17.1)

Table 2. Efficacy rate estimation of the Shingrix© and Zostavax© vaccines.

#RV3: Introduction: I think reference 26, Tsiu et al belongs in the Introduction to tell the reader about other surveys that have been done. Perhaps same for reference 25.

#AU3: We have included to the introduction the information provided by these references to previously conducted surveys (penultimate paragraph of the introduction): “Studies like Galvis V. et al. have already analyzed the importance of vaccination against HZ [20]. Surveys such as the one by Tsui E. et al. have been conducted to investigate the knowledge, attitudes and practice patterns of primary care physicians regarding the administration of herpes zoster [21].”

#RV4: Introduction lines 57-58: the statement “d) the introduction of vaccines for HZ (few reports on the influence of these vaccines on HZ rates have been published [8,11].” Consider modifying this statement or perhaps clarifying the intent of the statement, Several excellent publications in NEJM have described the remarkable efficacy of the Shingrix vaccine in various age groups.

#AU4: We have deleted our statement "few reports have been published on the influence of these vaccines on HZ rates".

#RV5: Introduction line 67: “Thus passive immunity measures such as vaccination are needed”. I suggest deleting “passive”. I do not think vaccination with attenuated live virus or subunit proteins is “passive” immunity.

#AU5: We have replaced the error: “Thus, active immunity measures such as vaccination are needed”.

#RV6: Some typos or grammar edits:

We have made all these proposed grammatical corrections.

  1. Line 159: change “through” to “thought”.

  1. Line 170: change “for vaccination” to “that vaccination”.

  1. Line 185: change “presuppose” to “had”.

  1. Lines 195-197: consider deleting “on a case-by-case basis, especially” from the sentence.

  1. Line 200: Consider changing “antibodies“ to “immunity” .... does not protect.

  1. Line 202: delete “we”.

  1. Line 241: Consider changing “This study” to “Our study”.

  1. Line 242: Place a period, not a comma, after “appropriately”.

Reviewer 3 Report

Please see my comments in the attached review.

Author Response

Reviewer 3

Review of Vaccines-2205576:  Vaccination against Varicella Herpes Zoster in the Survey of a Sample among Spanish Ophthalmologic Community.

Synopsis

Varicella-zoster virus is a human tropic alphaherpesvirus that causes a primary disease, chicken pox, upon infection but can recrudesce and cause shingles/herpes zoster (HZ) after a period of latency.  In some cases, there can be ophthalmic involvement.  Two vaccines for HZ are available, the live attenuated Zostavax and the gE-based subunit vaccine, Shingrix. This manuscript describes the results of a survey designed to ‘evaluate the knowledge of varicella-zoster virus (VZV) vaccines among ophthalmologists’ in Spain by proposing the question is it known among Spanish ophthalmologists/trainees that the population vaccinated against VZV has a lower rate of ophthalmological complications compared to the unvaccinated population.  A total of 206 of Spain’s ophthalmology professionals responded to the questionnaire demonstrating an association between professional experience and a reduction in the belief that VZV was attributed to vision loss.  Although most respondents (73%) were aware that vaccines for HZ are available knowledge about the significant superiority of Shingrix over Zostavax was stated to be limited.  The authors conclude that vaccination is effective at reducing HZ and that ophthalmologists should take an active role in recommending vaccines to reduce disease.

General comments

For the most part, this concise manuscript is well written.  However, there are numerous grammatical errors, leading to some confusion about how the data and subsequent discussion should be interpreted.  Moreover, this study appears to be very limited as there were only 206 participants in this study; there are more than 3,000 ophthalmologists in Spain.  The authors divide the participants into subspecialities but it is unclear what percentage these represent on a national level.  The limited number of participants becomes problematic when interpreting some outcomes.  This is clearly demonstrated on lines 177 to 178 where the authors state “…neither professional experience nor the development of the activity in a public or private center was a differentiating factor.”; the frequencies in each category are two small to make this statement.  In addition, it is also impossible to determine the accuracy of the authors findings as all 16 questions used for the questionnaire and the associated data are not provided by the authors.  There is also the possibility of sampling bias introduced by the way the questionnaire was designed and distributed, e-mail and WhatsApp, which was acknowledged by the authors in the discussion.  Although the sampling approach is common, it does not make it the correct approach.  Finally, did the survey answer the question, is it known among Spanish ophthalmologists/trainees that the population vaccinated against VZV has a lower rate of ophthalmological complications compared to the unvaccinated population?  Due to the studies limitations, it’s not clear whether the manuscript provides sufficient accurate information for the wider ophthalmology and VZV communities to answer in the affirmative.

Specific comments

Please address all grammatical errors throughout the manuscript.  Specific examples can be found on lines 34, 148, 202 to 204, and 207 to 208.  But, most importantly, the final sentence of the introduction, lines 79 to 80.

Line 100; Please provide all 16 questions and the frequency of responses for each question.

AU: We have added the survey with the 16 questions as an annex to the manuscript. All respondents answered all the questions (100%).

Lines 133 to 136; It would be useful to know what proportion of ophthalmologists were represented from each region.

AU: Unfortunately, we do not have the necessary data to perform this analysis. We know the number of participants from each region but we do not know what proportion they represent.

Lines 143 to 144; Did sampling bias account for the 50% of participants from public centers?  It would be useful to know what proportion of Spain’s ophthalmologists practice in public versus private practice.

AU: It is difficult to know if the distribution of ophthalmologists in public and private centers in our study is representative of the distribution in Spain. We have not found global data in this regard. However, we considered it interesting to investigate this percentage in our sample to analyze possible differences between public and private practice.

Line 157; “The greater the professional experience…” this phrase is not clear.  Please clarify.

We have incorporated Table 1 in order to improve the understanding of these results.

               Years of experience

<5

5-10

10-20

>20

p valor

Percentage of ophthalmologists who considered that VZV is a common cause of visual loss. n (%)

29 (74%)

18 (58%)

21(51%)

28(43%)

p=0.003

Table 1. Percentage of ophthalmologists who consider that VZV is common cause of visual loss according to the years of professional experience.

Lines 169 to 170; It is not clear whether the 26.7% of the professionals were surveyed or all of the professionals were surveyed.  The grammar is also poor.  Please clarify.

AU: We have made the following clarification: “26.7% of all respondents were unaware of the existence of vaccines”.

Lines 174 to 179; Given the very limited amount of data, is the section about vaccine recommendation worth dividing into specialty?  It might be best to remove as the statistical power is absent.

AU: Agreed, we have removed this information.

Lines 184 to 187; Why were only the Cornea and Ocular surface group specialties chosen?  These data do not seem relevant, especially given that a significant difference was not noted.

AU: We chose to include the results for the Cornea and Ocular surface subgroup since it is the subspecialty where VZV cases are seen most frequently, so we decided to study whether or not their knowledge about vaccination differed from the rest of the sample.

Lines 203 to 204; Why was it remarkable that most ophthalmologists would recommend vaccination if they were proven safe and effective?  Please clarify.

AU: We found remarkable that 93% would recommend vaccination and that only 4.3% of those surveyed had indicated the vaccine to their patients. We clarify it in the following sentence: “It is remarkable that most ophthalmologists that answered the questionnaire (93%) would recommend vaccination if it was proved to be safe and effective, but very few had prescribed it in their practice (4.3%).”

Lines 208 to 215; the authors state “…ophthalmologists may be less aware of the need and effectiveness of the HZ vaccine than general practitioners.”, yet most ophthalmologists (93%) from the authors study would recommend the HZ vaccine compared to 91% of 138 primary care physicians from a US institution.  It’s not clear what the authors are trying to achieve with this comparison or the message they are attempting to convey.  Please clarify.

AU: Thank you for your comment. We agree, this phrase is misleading, we have clarified it as follows: “As highlighted by Virgilio Galvis and colleagues, ophthalmologists may be less aware of the need and effectiveness of the HZ vaccine than general practitioners. [20] Although if we compare the results of our study with data from a survey of 138 physicians at an American institution (New York University Langone Health), we can see that the results in both specialties are similar. In general, 91% of primary care physicians recommended that all immunocompetent patients 60 years of age or older should be vaccinated, compared to 93% of ophthalmologists who answered the survey in our country and would recommend vaccination”.

Lines 254 to 255; this conclusion needs to be clarified to reflect the outcomes of the study.  For example, do ophthalmologists in Spain need appropriate training in recommending HZ vaccination etc.  Moreover, did they know that the population vaccinated against VZV has a lower rate of ophthalmological complications compared to the unvaccinated population?

AU: We have changed the conclusion as follows: “In conclusion, we have detected among the ophthalmologists surveyed that there is a lack of knowledge about the effectiveness of vaccination against VZV in the prevention of ophthalmological complications. In addition to low rates of indication and knowledge on how to request the vaccine. Higher vaccination rates in the near future will only be possible with the awareness and action of all physicians, both primary care, and specialists. VZV vaccine recommendation training programs could be carried out in Spain. We truly believe that the time has come for ophthalmologists to take an active role in HZO prevention.”

Round 2

Reviewer 1 Report

The manuscript is revised correctly alongside reviewr's comment. Tables are  not properly displayed?

Author Response

Reviewer 1 – Second Round

#RV0: The manuscript is revised correctly alongside reviewer’s comment.

#AU0: Thank you very much for your positive response.

#RV1: Tables are  not properly displayed?

#AU1: Thank you for the comment, the tables were with wrong format within the track version, we have solved it and also attached a clean version of the manuscript to improve the visual appearance.

Reviewer 3 Report

Thank you for addressing my comments.

Author Response

Reviewer 3 – Second Round

#RV0: Thank you for addressing my comments.

#AU0: Thank you very much for your positive response.